# Association between Sagittal Cervical Spinal Alignment and Degenerative Cervical Spondylosis: A Retrospective Study Using a New Scoring System

**DOI:** 10.3390/jcm11071772

**Published:** 2022-03-23

**Authors:** Chahyun Oh, Minwook Lee, Boohwi Hong, Byong-Sop Song, Sangwon Yun, Sanghun Kwon, Youngkwon Ko, Sun Yeul Lee, Chan Noh

**Affiliations:** 1Department of Anesthesiology and Pain Medicine, Chungnam National University Hospital, Daejeon 35015, Korea; ohchahyun@gmail.com (C.O.); koho0127@gmail.com (B.H.); yfreedom03@naver.com (S.Y.); green_sky@naver.com (S.K.); annn8432@gmail.com (Y.K.); 2Department of Anesthesiology and Pain Medicine, College of Medicine, Chungnam National University, Daejeon 34134, Korea; 3Department of Radiology, Yongin Severance Hospital, Yonsei University College of Medicine, Yongin 16995, Korea; eterna0204@gmail.com; 4Big Data Center, Biomedical Research Institute, Chungnam National University Hospital, Daejeon 35015, Korea; 5Core Laboratory of Translational Research, Biomedical Convergence Research Center, Chungnam National University Hospital, Daejeon 35015, Korea; bssongstat@gmail.com

**Keywords:** cervical spine, degenerative cervical spondylosis, cervical alignment, absolute rotational angle, sagittal vertical axis, kyphosis, lordosis

## Abstract

(1) Background: Prolonged neck flexion is thought to cause harmful loading on the cervical spine. Along with the degenerative process, cervical alignment tends to change toward lordotic curvature. The association between cervical alignment and cervical spondylosis remains unclear. (2) Methods: Three raters retrospectively assessed cervical radiographies of outpatients at a tertiary center in 2019 using degenerative cervical spondylosis score (DCS score; a newly developed scoring system), C2-7 absolute rotational angle (ARA), and C2-7 sagittal vertical axis (SVA). (3) Results: A total of 561 patients were included in the analysis. Multiple regression analysis with adjustments for age and sex revealed that C2-7 ARA, rather than SVA, was a significant parameter for degenerative spondylosis. The interaction between age and C2-7 ARA was significant, indicating that the increase in DCS score with increasing age was more pronounced in patients with kyphotic cervical alignment. The direct effect of age on DCS score was 0.349 (95% CI 0.319 to 0.380, *p* < 0.001) and the proportion of the mediation effect of C2-7 ARA was −0.125 (*p* < 0.001). (4) Conclusions: C2-7 ARA was significantly associated with DCS after adjustment for both age and sex. Subjects with more kyphotic cervical alignment showed a greater correlation between increased DCS score and older age.

## 1. Introduction

Degenerative cervical spondylosis is a chronic, acquired deterioration of the cervical spine that can cause neck pain, radiculopathy, and/or myelopathy. Although it can occur as early as the first decade of life, aging is associated with degenerative changes of the spine [1]. Loss of height and desiccation of the disc predispose its annulus to fissure and can cause herniation of the nucleus pulposus. Foraminal or spinal canal narrowing, sensitization of nociceptive fibers, and degenerative changes of facet joints occur dependently or independently of the disc degeneration and can be manifested as radiologic changes and/or clinical symptoms [2]. Along with these degenerative processes, cervical sagittal alignment tends to change toward lordotic curvature [3,4,5,6]. 

Excess use of mobile devices has recently been reported to be associated with increases in degenerative changes of the cervical spine and neck complaints [7,8]. Because a flexed neck posture, which is commonly observed during the use of hand-held mobile devices [9,10,11], can increase cervical loading [12], cervical malalignment (e.g., loss of lordosis or kyphosis) is likely associated with cervical spondylosis. 

To date, however, the association between degenerative cervical spondylosis and sagittal cervical alignment remains unclear, with few studies assessing their relationship. In this study, therefore, degenerative cervical spondylosis was assessed quantitatively using a newly developed scoring system, and the impact of cervical alignment on the degenerative score was assessed using an interaction model and a causal mediation effect model.

## 2. Materials and Methods

### 2.1. Study Design and Population

This retrospective cohort study included outpatients who underwent a set of cervical radiography examinations (six standard images, including antero-posterior, lateral, right and left oblique, flexion and extension views) during 2019 at Chungnam National University Hospital, a tertiary teaching hospital. The protocol of this study was approved by the Institutional Review Board of Chungnam National University Hospital (CNUH 2021-03-036). Patients were excluded if they had undergone surgical procedures or sustained fractures of the cervical spine; had anatomical abnormalities that could significantly distort alignment (e.g., congenital vertebral fusion); had a bone or soft tissue tumor in the cervical region, diffuse idiopathic skeletal hyperostosis (DISH), ossification of the posterior longitudinal ligament (OPLL), or ankylosing spondylitis. Patients with images in which the sagittal cervical alignment could not be assessed were also excluded, including images with ill-defined bony margins or when the image of the seventh cervical spine was blocked by the shoulder. Clinical data, including age, sex, and the department ordering the imaging study, were also recorded. Radiographic images were independently assessed by three authors (C.O., an anesthesiology fellow; M.L., a clinical professor of musculoskeletal radiology at a secondary hospital; and C.N., a clinical professor of pain medicine at a tertiary hospital) using the basic tools of the Picture Archiving and Communicating System (PACS; Maroview, Marosis, Korea). One month after the initial image assessment, images from 32 patients, chosen by stratified randomly sampling based on their initial score, were again assessed to evaluate intra-rater agreement. This manuscript adheres to the applicable STROBE (Strengthening the Reporting of Observational Studies in Epidemiology) guidelines [13]. 

### 2.2. Scoring System (Degenerative Cervical Spondylosis [DCS] Score)

The scoring system was developed using images from 30 randomly selected patients. Initially, the three authors designated to assess the images used a previously developed scoring system, the cervical degenerative index [14]. All authors subsequently discussed the results of inter- and intra-rater agreement and the limitations of the cervical degenerative index. This index was subsequently modified by: (1) adding a separate item for the facet joint; (2) separately assessing posterior and anterior osteophytes; (3) grading of anterior osteophytes as none, <2 mm, or ≥2 mm; (4) grading endplate sclerosis as none, detectable, or definite; and (5) grading of listhesis as absent, none or present. Modifications (1) and (2) were performed to improve discriminating ability, whereas modifications (3), (4) and (5) were performed to reduce inter-rater disagreement and to consider clinical relevance. The final newly developed scoring system, called the degenerative cervical spondylosis (DCS) score, is shown in Table 1. Each segment (C2-7 to C6-7) was rated separately and their scores added, resulting in a possible total score ranging from 0 to 60. A set of sample images and corresponding scores determined by the radiologist (M.L.) are presented in Appendix A. 

### 2.3. Assessment of Sagittal Cervical Alignment

Cervical alignment on lateral images was assessed using two methods (Figure 1). In the first method, C2-7 absolute rotational angle (ARA; °) [15] was determined by drawing two lines, parallel to the posterior margins of the vertebral bodies of C2 and C7, and measuring the angle between these two lines. A negative angle represents lordotic curvature. In the second method, C2-7 sagittal vertical axis (SVA; mm) [16], a vertical line was drawn from the center of the vertebral body of C2 to the ground. The vertical distance from this line to the posterior superior aspect of the C7 vertebral body was measured. 

C2-7 absolute rotational angle (ARA): angle between the two lines, one parallel to the posterior margin of the vertebral bodies of C2 and the other parallel to the posterior margin of the vertebral bodies of C7 (orange solid lines). A negative angle represents lordotic curvature. C2-7 sagittal vertical axis (SVA): vertical distance from the C2 plumb line (red dotted line) to the posterior superior aspect of the C7 vertebral body (red solid line). 

### 2.4. Statistical Analyses

Sample size was based on the data available from January to December 2019. No statistical power calculation was performed before the study. A two-way mixed effect model was used to calculate intra- and inter-rater correlation coefficients, with absolute agreement assessed for intra-rater correlation and consistency for inter-rater correlation [17]. The mean values of the measurements made by the three raters were used for the analysis, except for the analysis of intra- and inter-rater correlations. Continuous variables are presented as mean and standard deviation (SD) or as median and interquartile ranges (IQR) after testing for normal distribution using the Shapiro–Wilk test. The measured values were stratified by sex, and, depending on their distribution, the differences between men and women were assessed by independent t-tests or Mann–Whitney U tests. 

Correlations between the continuous variables of interest (age, DCS score, C2-7 ARA, C2-7 SVA) were assessed by Spearman correlation analysis. A linear regression analysis (model 1) was performed using DCS score as a dependent variable and C2-7 ARA and C2-7 SVA as independent variables, with adjustment for confounding factors (age and sex). Additionally, to assess whether cervical alignment, which had been shown significant on previous analysis, had a moderating effect on the relationships of DCS score with age and sex, interaction terms between these confounders and cervical alignment were tested (model 2).

Finally, to determine whether C2-7 ARA has any mediating effect on the degenerative process of the cervical spine (i.e., age vs. DCS score), a causal mediation effect model, which considered sex and C2-7 SVA as covariates, was applied. The main concept of the mediation theory is that a mediating variable is causally related to the independent and dependent variables, such that changing the mediating variable will change the response variable [18]. Average causal mediation effect (ACME) and average direct effect (ADE), which represent the population averages of these causal mediation and direct effects, respectively, were calculated. Significance tests for mediation effect were calculated by computing the nonparametric boot-strap confidence interval. *p*-values < 0.05 were defined as statistically significant. All statistical analyses were performed using R software version 4.0.4 (R Project for Statistical Computing, Vienna, Austria), with the “mediation” package [19] used for the mediation analysis. 

## 3. Results

Of the 699 patients included in the initial feasibility screening, 108 were excluded. Of the remaining 591 patients, 30 were used in the development of the scoring system, and 561 were included in the final analysis (Figure 2). The clinical characteristics and mean measured values of these patients, stratified by sex, are shown in Table 2. Median [IQR] DCS scores (11.3 [6.0–18.0] vs. 8.3 [4.0–13.3], *p* < 0.001) and mean ± SD C2-7 SVA (21.7 ± 10.9 mm vs. 15.5 ± 9.3 mm, *p* < 0.001) were significantly higher in men than in women.

The inter-rater correlation coefficient for DCS score was 0.87 (95% CI 0.85 to 0.88), indicating good reliability [17]. The inter-rater correlation coefficients were 0.94 for both C2-7 ARA and SVA, indicating excellent reliability (Appendix A). Intra-rater correlation coefficients for DCS score and C2-7 ARA and SVA also showed excellent reliability (Appendix A). 

Correlations between the continuous variables of interest (age, DCS score, and C2-7 ARA and SVA) are summarized in Table 3. Age showed significant positive correlations with all three measurements. The correlation between DCS score and C2-7 ARA was not significant, whereas the correlation between DCS score and C2-7 SVA was poor, but statistically significant (rho = 0.182, *p* < 0.001). The regression analysis assessing relationship between DCS score and cervical alignment with adjustments for age and sex (model 1) showed that C2-7 ARA (*p* < 0.001), rather than SVA (*p* = 0.136), is significantly associated with DCS score (Table 4). The interaction between age and C2-7 ARA was also significant (model 2). The increase in DCS score with increasing C2-7 ARA was more pronounced in the older population (Figure 3), and, conversely, the increase in DCS score with age was more pronounced in patients with kyphotic cervical alignment (Figure 4).

A negative angle represents lordotic curvature. The increasing trend of DCS score with increasing C2-7 ARA was more pronounced in the older population.

A negative angle represents lordotic curvature. The increasing trend of DCS score with increasing age was more pronounced in patients with kyphotic cervical alignment.

The direct effect (ADE) of age on DCS score was 0.349 (95% CI 0.319 to 0.380, *p* < 0.001) and the mediation effect (ACME) of C2-7 ARA was −0.039 (95% CI −0.056 to −0.030, *p* < 0.001). The total effect (i.e., the sum of direct and indirect effects) was 0.308 (95% CI 0.280 to 0.340, *p* < 0.001) and the proportion of the mediation was −0.125 (95% CI −0.188 to −0.080, *p* < 0.001) (Appendix A).

## 4. Discussion

Degenerative changes of the cervical spine with increased age are intuitive and previously reported [20,21,22]. Cervical alignment also changes with age [3], while differing by sex [23]. A previous study found that the correlation between cervical lordosis and degenerative change was not statistically significant, perhaps because these confounding factors were not considered [24]. The present study showed a significant association between DCS and C2-7 ARA after adjustment for age, sex, and C2-7 SVA. Although C2-7 SVA was positively correlated with DCS score, this correlation lost its statistical significance when the other confounders are collectively considered. As the other factors cannot be considered separately in reality, the multivariate models suggested in the current study would provide a more clinically meaningful conclusion. Additionally, C2-7 ARA had a significant moderating effect on the association between age and DCS score. That is, in subjects with a more kyphotic cervical alignment, DCS score tended to increase more steeply with increasing age. Mediation analysis found that (1) the ratio of 0.349 in observed degenerative changes of the cervical spine (DCS score) in this population was due to age; (2) C2-7 ARA had an effect on DCS score opposite to that of age (−0.039; 12.5% of the total effect); and (3) the total effect of 0.308 is the sum of the direct effect of age and the indirect (mediation) effect of C2-7 ARA.

The prevalence of cervical pathologies Is increasing with increased use of mobile devices [7,8,25] and computers, which are usually associated with prolonged flexed neck posture [9,10,11,12]. In addition, prolonged neck flexion has been reported to cause harmful loading on the cervical spine [12,26,27], and thus may accelerate degenerative processes in the cervical spine. Loss of normal lordotic curvature has been recently reported to increase in the Korean population, especially in younger individuals [28]. Furthermore, recent studies indicate that cervical alignment is associated with postoperative outcome [29] and disc herniation [30], and an emerging trend of increasing interest for cervical alignment was noted [31]. In this context, the findings of the current study, showing a significant relationship between cervical alignment and DCS score, warrant further clinical research. 

The causal mediation effect model applied in the present study revealed that sagittal cervical alignment had a mediation effect on degenerative changes of the cervical spine with increased age. Lordotic changes in cervical alignment may have a protective effect against degenerative processes of the cervical spine. However, as this analysis was based on cross-sectional data, with all measurements in each subject performed at the same time, caution should be exercised in interpreting these results. Although mediation analysis has been performed frequently in various scientific contexts [32] and has a sound statistical background [18], the interpretation of its results is not completely free from clinical inferences or temporal relationships between the variables [32,33,34]. Thus, the mediation effect revealed in the present study should be considered hypothesis generating. Future studies with a longitudinal or interventional design are warranted to confirm whether preservation or promotion of a lordotic curvature has clinical benefits in preventing degenerative processes of the cervical spine. 

Although there was a pre-existing quantitative scoring system for DCS, the cervical degenerative index, it has limited intra- and inter-rater correlation and could not discretely evaluate the facet joint. Thus, a new scoring system was developed for the present study. To minimize ambiguity and intra- or inter-rater disagreement in the assessment of sclerosis, osteophytes, and listhesis, the system was simplified by reducing levels of classification in each category. Furthermore, to improve clinical relevance, posterior osteophytes and facet joints were also evaluated. These added categories are considered important in cervical pathologic conditions, such as neural foraminal stenosis [35] and facet joint syndrome [36]. The excellent intra- and inter-rater agreements shown in the current study provide reliability of the analysis and can firmly support the conclusion. Additionally, similar findings by the three raters—a radiologist, a pain physician, and an anesthesiologist—despite differences in their expertise, suggest the generalizability of the newly developed scoring system. 

Cervical radiography used in the current study is clinically useful, despite the availability of more sophisticated imaging methods, such as magnetic resonance imaging. Using simple and low-cost images, cervical radiography can assess cervical alignment in the neutral or flexion/extension position and can visualize overall degenerative changes. Thus, cervical radiography is still considered a first-line imaging modality for patients with cervical pathologies. 

This study had several limitations. First, due to its cross-sectional design, the causal relationship between cervical alignment and DCS could not be determined, suggesting a need for additional, confirmatory studies. Second, the newly developed scoring system has not yet been validated externally and its clinical relevance is not known. Third, the excellent inter-rater agreement observed in the current study may have resulted from the discussion held during the development stage of the scoring system. External validation is therefore needed to confirm the inter-rater agreement. 

## 5. Conclusions

In conclusion, C2-7 ARA was significantly associated with DCS score after adjustment for both age and sex. In patients with a more kyphotic cervical alignment, a greater increase in DCS score was observed as age increased. Further studies are warranted to evaluate whether lordotic cervical alignment has a protective effect against the degenerative process of the cervical spine.

## Figures and Tables

**Figure 1 jcm-11-01772-f001:**
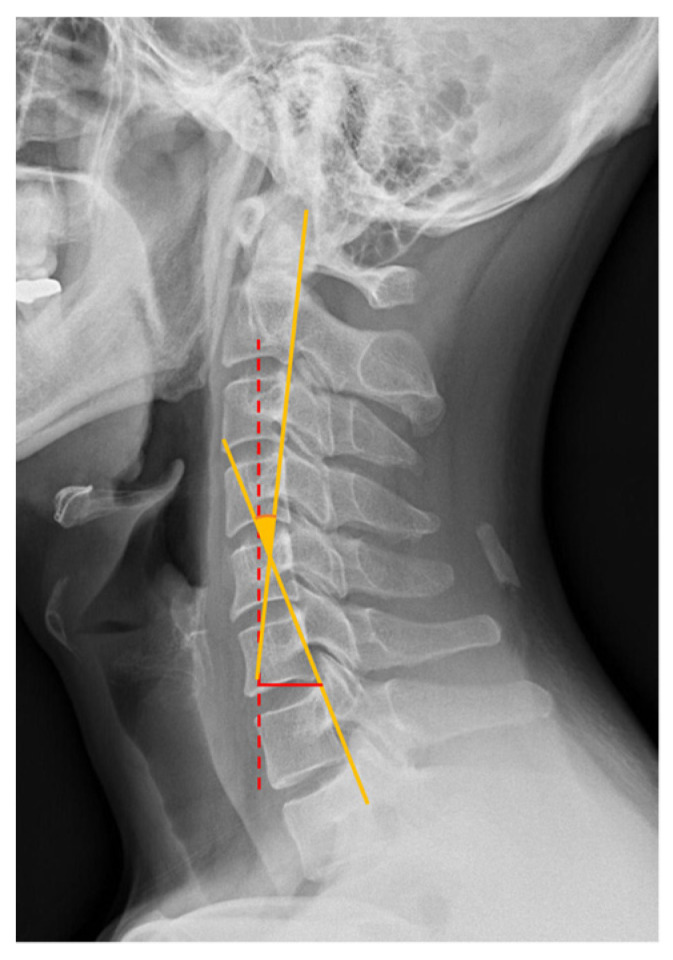
Radiographic measurement of sagittal cervical alignment.

**Figure 2 jcm-11-01772-f002:**
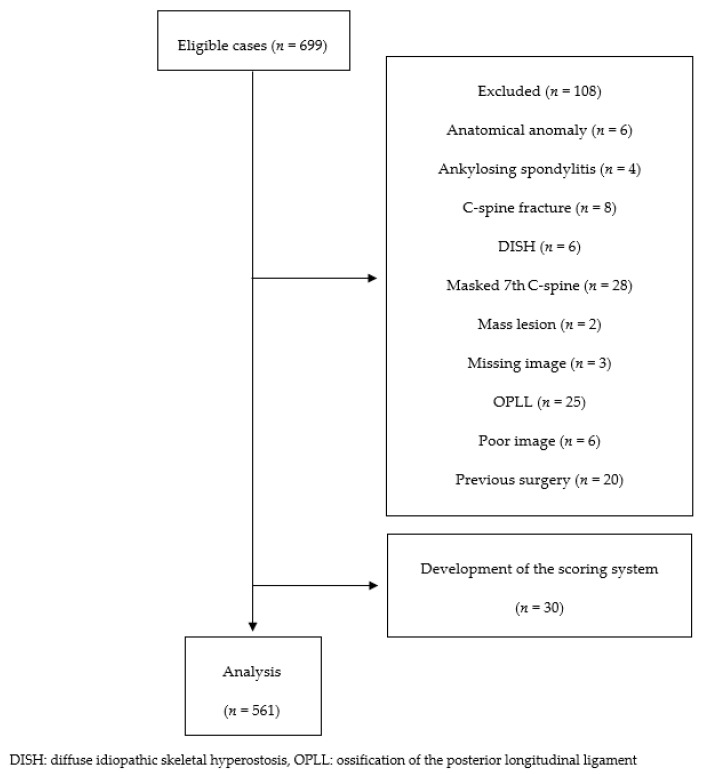
Patient flow diagram.

**Figure 3 jcm-11-01772-f003:**
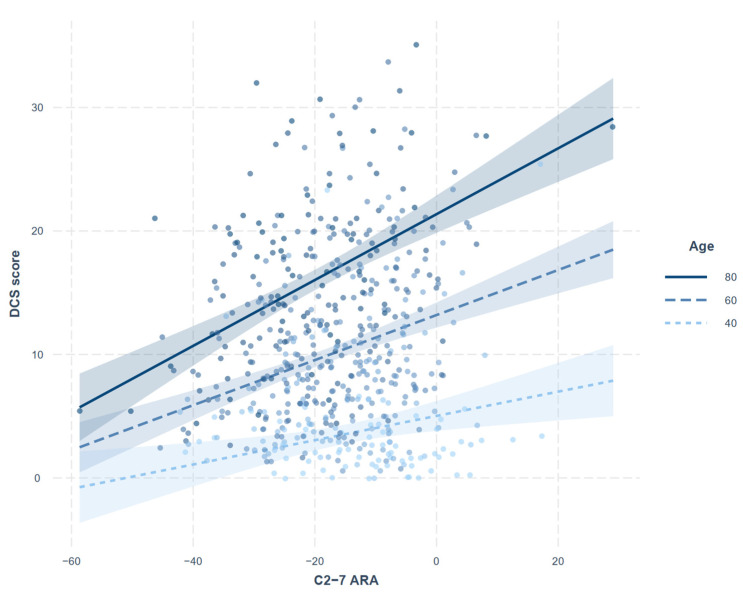
Interaction plot showing the moderating effect of age on the relationship between degenerative cervical spondylosis (DCS) score and C2-7 absolute rotational angle (ARA).

**Figure 4 jcm-11-01772-f004:**
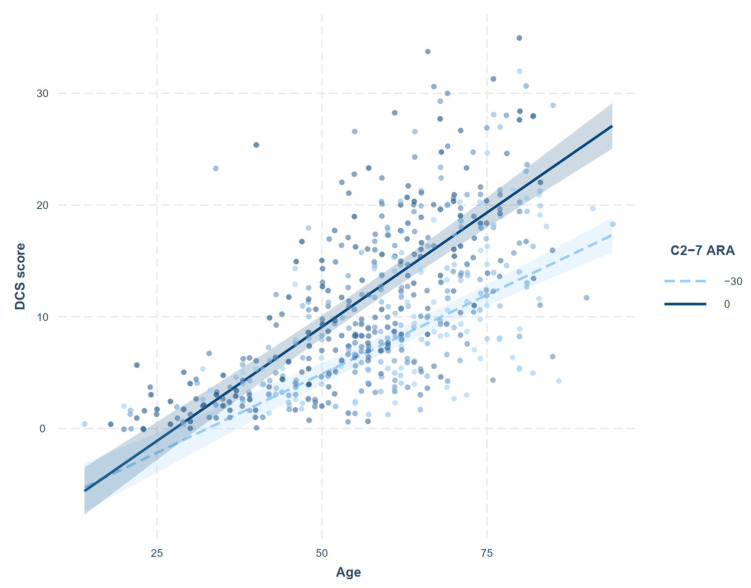
Interaction plot showing the impact of C2-7 absolute rotational angle (ARA) on the relationship between degenerative cervical spondylosis (DCS) score and age.

**Table 1 jcm-11-01772-t001:** Description of the newly developed degenerative cervical spondylosis score (DCSS).

Disc Degeneration		
Sclerosis	None	0
	Detectable	1
	Definite (with irregularity)	2
Narrowing ^1^	<25%	0
	25~50%	1
	51~75%	2
	>75%	3
Anterior osteophytes	None	0
	<2 mm	1
	≥2 mm	2
Posterior osteophytes ^2,3^	None	0
	<50%	1
	≥50%	2
Listhesis ^4^	None	0
	Present	1
Facet joint		
	None or mild sclerotic change	0
	Mild narrowing or irregularity	1
	Severe narrowing	2

^1^ Middle disc height measured in a lateral image compared with a normal (or least degenerated) disc height at any segment of the cervical spine; the mid-point of the margins was selected in segments with ill-defined or sub-optimally aligned endplates. ^2^ Osteophytes originating from the vertebral body or intervertebral disc (not the facet joint). ^3^ The amount of blockage of the intervertebral foramen by posterior osteophytes in oblique images. The left and right sides of the segment were compared, and the side with the higher grade was selected. ^4^ Either retro- or antero-listhesis.

**Table 2 jcm-11-01772-t002:** Clinical characteristics and mean measured values of patients stratified by sex.

Sex	Female	Male	*p*
	(*n* = 317)	(*n* = 244)	
Age (yr), median [IQR]	60.0 [49.0–69.0]	58.0 [46.0–68.5]	0.222
Department, *n* (%)			0.147
Anesthesiology and pain medicine	23 (7.3)	15 (6.1)	
Neurology	42 (3.2)	25 (10.2)	
Neurosurgery	49 (5.5)	53 (21.7)	
Orthopedic	165 (52.1)	130 (53.3)	
Rheumatology	19 (6.0)	6 (2.5)	
Rehabilitation medicine	19 (6.0)	15 (6.1)	
DCS score, median [IQR]			
Endplate sclerosis	1.7 [1.0–3.0]	2.3 [1.0–3.7]	0.003
Disc space narrowing	0.7 [0.0–1.7]	1.0 [0.0–2.3]	0.036
Anterior osteophyte	1.3 [0.3–3.0]	2.3 [0.7–4.3]	<0.001
Posterior osteophyte	2.0 [1.0–3.0]	2.8 [1.7–4.0]	<0.001
Listhesis	1.0 [0.3–1.7]	1.0 [0.7–2.0]	0.012
Facet joint	0.7 [0.3–1.7]	1.3 [0.3–2.7]	<0.001
Total score	8.3 [4.0–13.3]	11.3 [6.0–18.0]	<0.001
Sagittal alignment, mean ± SD			
C2-7 ARA (°)	−17.5 ± 11.9	−16.2 ± 11.0	0.179
C2-7 SVA (mm)	15.5 ± 9.3	21.7 ± 10.9	<0.001

DCS: degenerative cervical spondylosis, ARA: absolute rotational angle, SVA: sagittal vertical axis.

**Table 3 jcm-11-01772-t003:** Correlations between age, cervical alignment, and degenerative cervical spondylosis score.

	Age	C2-7ARA	C2-7SVA
C2-7ARA	−0.25 *		
C2-7SVA	0.14 *	0.22 *	
DCS score	0.66 *	0.04	0.18 *

ARA: absolute rotational angle, SVA: sagittal vertical axis, DCS: degenerative cervical spondylosis. * *p* < 0.001.

**Table 4 jcm-11-01772-t004:** Summary of the multiple linear regression models for degenerative cervical spondylosis score.

	Model 1		Model 2	
	Estimate	*p*	Estimate	*p*
(Intercept)	−7.472		−11.318	
Male	3.526	<0.001	2.622	0.001
Age	0.346	<0.001	0.408	<0.001
C2-7ARA	0.168	<0.001	0.070	0.318
C2-7SVA	−0.034	0.136	NA	NA
C2-7ARA * Age	NA	NA	0.004	<0.001
C2-7ARA * Male	NA	NA	−0.041	0.284

Model 1: linear model (adjusted R^2^ = 0.506), Model 2: linear model with interaction terms (adjusted R^2^ = 517). ARA: absolute rotational angle, SVA: sagittal vertical axis, NA: not available. The asterisks in the table indicate interaction between variables.

## Data Availability

The data presented in this study are available on reasonable request from the corresponding author.

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
