# Peer review of "Association between Sagittal Cervical Spinal Alignment and Degenerative Cervical Spondylosis: A Retrospective Study Using a New Scoring System"

_jcm, 2022, doi:10.3390/jcm11071772_

Round 1
Reviewer 1 Report
First of all thank you very much for the work of the author. As a newly developed scoring system, DCS provided a detailed quantitative radiographic assessment of spondylitic change per cervical spine level. The authors found the significant association between C2-7 ARA and modified DCS, well-structured and articulate. But some areas need to be improved. This study revealed that C2-7 ARA rather than SVA was a significant parameter for degenerative spondylosis. However, I couldn’t find detailed explanation of this finding. In Deed E’s comparative study of Cobb method or Harrison posterior tangent method, he suggested that the Harrison posterior tangent method is superior to the Cobb method on which to draw. There is a lack of literature in the past three years in the references, and it is recommended to supplement. Moreover, I suggested that authors could make a brief literature review of degenerative cervical spondylosis’s natural history. I have a recommendation for additional citations: A Bibliometric Analysis and Visualization of Current Research Trends in the Treatment of Cervical Spondylotic Myelopathy. I hope the authors could further improve this study in the above aspects.
Author Response
Reviewer #1
First of all thank you very much for the work of the author. As a newly developed scoring system, DCS provided a detailed quantitative radiographic assessment of spondylitic change per cervical spine level. The authors found the significant association between C2-7 ARA and modified DCS, well-structured and articulate. But some areas need to be improved.
- This study revealed that C2-7 ARA rather than SVA was a significant parameter for degenerative spondylosis. However, I couldn’t find detailed explanation of this finding.
Thank you for the comment. After careful review of our previous description regarding this issue, we found that there is some need for more elaboration.
As described in the statistics section, we fitted a linear model which assessing relationship between DCS score (dependent variable) and cervical alignment (independent variable) with adjusting age and sex as covariates.
So, the model 1 was
DCS score = k1C2-7 ARA + k2C2-7 SVA + k3Age + k4Sex
(k1 to 4 are coefficients)
In this model, C2-7 ARA rather than C2-7 SVA was the factor revealed to be statistically significant.
To address these meanings more clearly, we revised the description as follows:
“The regression analysis assessing relationship between DCS score and cervical alignment with adjustments for age and sex (model 1) showed that C2-7 ARA (p<0.001), rather than SVA (p=0.136), is significantly associated with DCS score (Table 4).”
- In Deed E’s comparative study of Cobb method or Harrison posterior tangent method, he suggested that the Harrison posterior tangent method is superior to the Cobb method on which to draw.
Thank you for the comment. As you pointed out, the Harrison posterior tangent method is the recommended method for the assessment of cervical alignment in reference to Deed E’s study. We also considered this issue to be important and therefore used C2-7 ARA in our study (reference #15). Although we found that there are some confusion or ambiguity in the terminology for posterior tangent method, the study by Deed E clearly stated that “Segmental angles derived from the posterior tangents are termed relative rotation angles (RRAs). The sum of these angles from RRAC2–C3 to RRA C6–C7 is termed the absolute rotation angle (ARAC2–C7) and is the angle between Jackson’s physiologic stress lines drawn at the C2 and C7 posterior body margins”.
In this sense, we considered C2-7 ARA (angle between Jackson’s physiologic stress lines) as equivalent as Harrison posterior tangent method (sum of these angles from RRAC2–C3 to RRA C6–C7) although there would be a methodological difference between them.
- There is a lack of literature in the past three years in the references, and it is recommended to supplement.
Thank you for the comment. We agree that reviewing more recent articles is helpful for the current manuscript. Following articles are reviewed and cited in the manuscript accordingly.
# Pinter, Z.W.; Salmons, H.I.t.; Townsley, S.E.; Xiong, A.; Michalopoulos, G.D.; El Sammak, S.; Currier, B.; Nassr, A.; Freedman, B.A.; Bydon, M.; et al. Improved Sagittal Alignment is Associated with Early Postoperative Neck Disability and Pain Related Patient Reported Outcomes following Posterior Cervical Decompression and Fusion for Myelopathy. World Neurosurg 2022, doi:10.1016/j.wneu.2022.02.075.
# Charles, Y.P.; Prost, S.; Pesenti, S.; Iharreborde, B.; Bauduin, E.; Laouissat, F.; Riouallon, G.; Wolff, S.; Challier, V.; Obeid, I.; et al. Variation of cervical sagittal alignment parameters according to gender, pelvic incidence and age. Eur Spine J 2022, doi:10.1007/s00586-021-07102-w.
# Gao, K.; Zhang, J.; Lai, J.; Liu, W.; Lyu, H.; Wu, Y.; Lin, Z.; Cao, Y. Correlation between cervical lordosis and cervical disc herniation in young patients with neck pain. Medicine 2019, 98, e16545, doi:10.1097/md.0000000000016545.
# Al-Hadidi, F.; Bsisu, I.; AlRyalat, S.A.; Al-Zu'bi, B.; Bsisu, R.; Hamdan, M.; Kanaan, T.; Yasin, M.; Samarah, O. Association between mobile phone use and neck pain in university students: A cross-sectional study using numeric rating scale for evaluation of neck pain. PLoS One 2019, 14, e0217231, doi:10.1371/journal.pone.0217231.
# Yin, M.; Xu, C.; Ma, J.; Ye, J.; Mo, W. A Bibliometric Analysis and Visualization of Current Research Trends in the Treatment of Cervical Spondylotic Myelopathy. Global Spine J 2021, 11, 988-998, doi:10.1177/2192568220948832
- Moreover, I suggested that authors could make a brief literature review of degenerative cervical spondylosis’s natural history.
The first paragraph of the manuscript was revised as follows:
“Degenerative cervical spondylosis is a chronic, acquired deterioration of the cervical spine that can cause neck pain, radiculopathy, and/or myelopathy. Although it can occur as early as the first decade of life, aging is associated with degenerative changes of the spine. Loss of height and desiccation of the disc predisposes its annulus to fissure and can cause herniation of the nucleus pulposus. Foraminal or spinal canal narrowing, sensitization of nociceptive fibers, and degenerative changes of facet joint occur dependently or independently of the disc degeneration and can be manifested as radiologic changes and/or clinical symptoms. Along with these degenerative processes, cervical sagittal alignment tends to change toward lordotic curvature.”
- I have a recommendation for additional citations: A Bibliometric Analysis and Visualization of Current Research Trends in the Treatment of Cervical Spondylotic Myelopathy. I hope the authors could further improve this study in the above aspects.
Thank you for the invaluable suggestion. We carefully reviewed the article you recommended and added to the citations accordingly.
“Also, recent studies indicate that cervical alignment is associated with postoperative outcome and disc herniation, and an emerging trend of increasing interest for cervical alignment was noted. In this context, the findings of the current study, showing a significant relationship between cervical alignment and DCS score, warrant further clinical research.”

Reviewer 2 Report
The authors described significant association between degenerative change of cervical spine and cervical alignment. The present manuscript includes important and interesting data. Before acceptance, the authors had better consider follwoing point.
The present data showed that SVA had no significant association with degenerative change. This means "SVA showed no deterioration altough alignment became worse"? If so, what factor compensate alignment deterioration?
Author Response
Reviewer #2
The authors described significant association between degenerative change of cervical spine and cervical alignment. The present manuscript includes important and interesting data. Before acceptance, the authors had better consider following point.
The present data showed that SVA had no significant association with degenerative change. This means "SVA showed no deterioration although alignment became worse"? If so, what factor compensate alignment deterioration?
Thank you for the invaluable comment.
First of all, we respectfully speculate that the reviewer actually meant “SVA showed no deterioration although DCS became worse (increase)?”. If this was the case, following response would be appropriate:
According to the result of our analysis, specifically from the regression analysis (model 1), SVA caused non-significant effect on DCS when the other factors including age, sex, and C2-7 ARA were collectively considered. According to this model, one could imagine such clinical scenario that DCS score can be preserved provided the spine maintains lordotic curvature (when age and sex are constant) regardless of whether SVA deteriorates (i.e. increases) or not. In the same manner, another clinical scenario is also possible that DCS score is increased provided the spine curvature is kyphotic regardless of whether SVA deteriorates or not. In this sense, it should be interpreted as “SVA was irrelevant” rather than “SVA showed no deterioration”.
In fact, C2-7 SVA and DCS score was positively correlated (Table 3). However, as the other factors such as age, sex, and C2-7 ARA cannot be considered separately in reality, a multivariate model such as model 1 would provide more clinically meaningful conclusion.
If the reviewer’s intend was actually "SVA showed no deterioration although alignment (C2-7 ARA) became worse?", then following response would be appropriate:
According to the result of our analysis, C2-7 SVA and C2-7 ARA showed a positive correlation. In other words, when C2-7 ARA increases (changing toward kyphotic curvature), C2-7 SVA would increase (deteriorate) too. Therefore, the directions of these two parameters were not conflicting, at least in the populational level. In this context, the meaning of the existence of a compensatory mechanism for SVA not to deteriorate during the other alignment parameter (C2-7 ARA) deteriorate is not clear.
Although we tried various combinations of variables to construct a model which could provide plausible mechanism for the potential compensatory mechanism (searching for a factor which has negative impact on the relationship between C2-7 SVA and C2-7 ARA), no such model could be derived within this study. Other parameters such as T1 slope or C0-2 angle may complement this issue. However, as the main theme of the current study was to assess relationship between cervical alignment and degenerative change, with all respect, we consider discussing this issue in detail would be beyond the scope of the current study.

Round 2
Reviewer 1 Report
My previous problems have been ended.
Author Response
Glad to hear that the response was appropriate.
Thank you for the review.